# An Investigation of Memorization Risk in Healthcare Foundation Models

**Sana Tonekaboni**[*]
MIT
Broad Institute of MIT and Harvard
Vector Institute
stonekab@mit.edu

**Lena Stempfle**[*]
MIT
Chalmers University of Technology
University of Gothenburg
stempfle@mit.edu

**Adibvafa Fallahpour**[*]
University of Toronto
Vector Institute
University Health Network (UHN)
adibvafa.fallahpour@mail.utoronto.ca

**Walter Gerych**
Worcester Polytechnic Institute
Computer Science Department
wgerych@wpi.edu

**Marzyeh Ghassemi**
MIT
mghassem@mit.edu

## Abstract

Foundation models trained on large-scale de-identified electronic health records (EHRs) hold promise for clinical applications. However, their capacity to memorize patient information raises important privacy concerns. In this work, we introduce a suite of black-box evaluation tests to assess privacy-related memorization risks in foundation models trained on structured EHR data. Our framework includes methods for probing memorization at both the embedding and generative levels, and aims to distinguish between model generalization and harmful memorization in clinically relevant settings. We contextualize memorization in terms of its potential to compromise patient privacy, particularly for vulnerable subgroups. We validate our approach on a publicly available EHR foundation model and release an open-source toolkit to facilitate reproducible and collaborative privacy assessments in healthcare AI. [2]

## 1   Introduction

Foundation models trained on Electronic Health Records (EHR-FM) have been proposed as a promising direction for advancing healthcare [38]. However, there are concerns that large models trained on sensitive patient data may memorize and expose private health information [36], even under legal de-identification. Previous work on memorization in Large Language Models (LLMs) found that targeted prompts can reveal private data from training data sets [11, 9, 5, 50, 30, 26]. For EHR-FM, the risk is comparable, but the sensitivity of clinical data makes it even more critical to *proactively* assess and mitigate memorization threats.

Model memorization is often studied through the lens of adversarial privacy attacks, such as membership inference, attribute inference, and training data reconstruction, that aim to extract sensitive

---

[*]Equal contribution.
[2]Code available at https://github.com/sanatonek/EHR-FM_memorization

39th Conference on Neural Information Processing Systems (NeurIPS 2025).

information from models [3, 40, 45, 19, 39]. While these approaches provide important lower bounds on leakage, they often ignore the deployment context of the model. Recent work has introduced the notion of contextual integrity [37], defining privacy breaches as violations of *context-specific norms* governing information flow. This distinction is particularly critical in clinical settings, where beneficial generalization of clinical knowledge may be similar to harmful memorization of patient-level information. We want a model to infer tuberculosis for a 36-year-old who presents with weight loss and cough. We do not want to infer tuberculosis for a 36-year-old who presents only with hypertension and knee pain, merely because that same patient had TB in the training data. In this work, we propose a contextual, risk-based evaluation framework for evaluating memorization in EHR-FM, with a focus on identifying practical privacy threats.

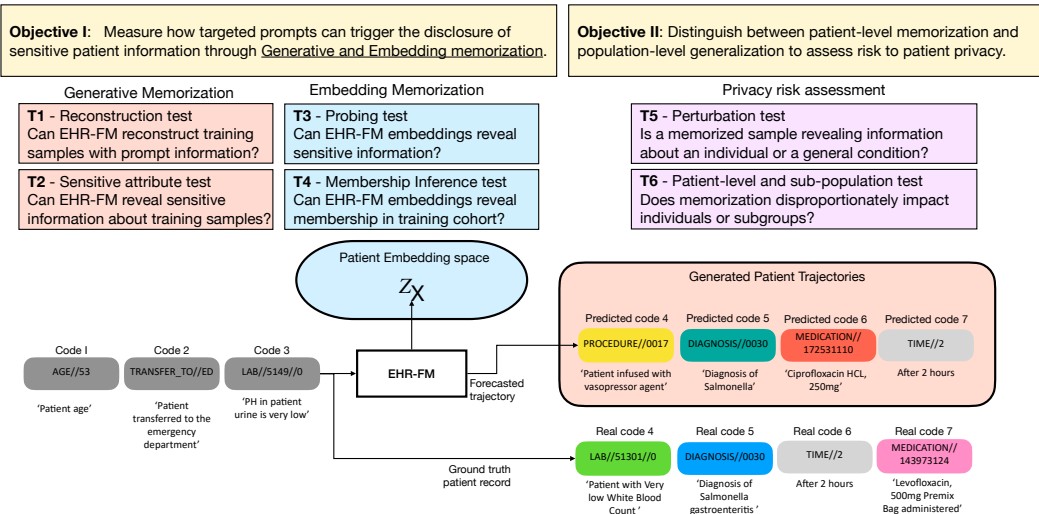

Figure 1: Our proposed tests (T1–T6) to evaluate memorization in EHR-FM, grouped by two core objectives. T1–T4 measure the extent to which models reveal training data, targeting both the embedding and the generated data domain. T5–T6 perform patient-level evaluations to quantify individual privacy risks from memorization or data leakage.

We explore memorization and its risk to patient privacy in EHR-FM trained on private data sequences of structured medical codes, and released as *black-boxes* with *prompt-only access*. EHR-FMs are distinct from clinical LLMs as they are not trained with free text such as clinical notes, where memorization risks have been explored [29]. We introduce a set of pragmatic EHR-FM tests with two main objectives (Figure 1): (1) to quantify different types of memorization that can be triggered via prompting, and (2) to assess the associated privacy risks by distinguishing harmful memorization at the patient level from benign population-level generalization. Overall, these tests evaluate the privacy risk of data leakage as a function of information an adversary can access. Our tests can be used to identify vulnerable samples and subcategories of individuals for further safeguarding prior to public release of a model using post-training safety layers [31], red-teaming [42] or retraining [28, 25].

We first introduce a practical formalization of memorization tailored to EHR-FMs. Next, we design and evaluate two **generative memorization tests** (T1–T2) on a publicly available EHR-FM [14] to assess its ability to reconstruct training data. Our results show that minimal individual-level knowledge (e.g., age) does not reveal sensitive attributes; however, we identify the additional contextual information that can trigger leakage. Our proposed **privacy risk assessment tests** (T5–T6) quantify patient-level memorization risk and assist in distinguishing between patient-level memorization and generalization. We also employ two **embedding memorization tests** (T3–T4), motivated by prior concerns that deep embeddings can leak private information [49]. These tests examine both membership inference and leakage of sensitive medical attributes from embeddings (overview in Figure 7 in the appendix). These tests address an important yet understudied aspect of EHR-FMs and offer a practical framework for systematically evaluating memorization and privacy risks as these models continue to scale and see broader clinical use.

## 2 Related work

**EHR Foundation Models**   EHR-FMs are large models trained on structural EHR data using self-supervision at scale that can be adapted to a wide range of downstream tasks [6]. These models process sequences of standardized medical codes (diagnosis ICD, procedures, medications, lab tests) alongside temporal tokens indicating time intervals, creating patient trajectories (example shown in Figure 1). Two main category of EHR-FM includes Encoder models, like BEHRT [32], CLMBR-T [52], and CEHR-BERT [41] for representation learning and Decoder models like ESGPT [35], ETHOS [44], EHRMamba [14] that are trained to generate patient records. Initial work in EHR-FM has demonstrated significant improvements over task-specific architectures in limited data settings [52], domain adaptation, and robustness to distribution shifts [17]. However, study of memorization in these models, or even a definition of what memorization is in this setting is still an open problem.

**Memorization in Privacy and Security**   Memorization has been extensively studied in privacy and security research [3, 40, 45, 19]. Categories of attacks include but are not limited to membership inference [48, 12, 20], property inference [49, 16], and model extraction [27], each targeting different aspects of models and assuming varying levels of transparency. Memorization in these settings is typically defined as being able to extract any training data from the model [11, 40]. However, in EHR settings, it is often more practical to assess the *sensitivity* level of the extracted data focusing on the deployment context [51].

**Memorization in Foundation Models**   The study of memorization of foundation models mainly focuses on large language models (LLMs) [11, 9, 10, 40]. Memorization is described as a model's capacity to store and reproduce specific training data patterns which can result in data leakage. While high-capacity models differ from standard ML models in their capacity to generalize without overfitting [11, 7], they are still vulnerable to data leakage [10, 54, 34, 9, 24] by membership inference [48, 53], property inference [16], and model inversion attacks [15].

**Memorization in Healthcare Foundation Models**   Memorization in domain-specific foundation models, particularly those trained on medical data, lacks comprehensive study. In particular, their implications for healthcare domains and patient privacy remain unclear. The closest work to ours investigates extraction of personal health identifiers from an LLM trained on medical data [29], finding basic probes ineffective while noting advanced attacks' potential. We focus on models trained on structural EHR data to investigate how such risks could manifest in rich temporal medical records.

## 3 Testing for Memorization

In EHR-FM systems, we expect models to learn from every patient they encounter and internalize that knowledge to make better predictions. Generalization is desirable, but model should not reveal memorized examples if such disclosures compromise patient privacy. To address this, we introduce a set of practical evaluation tests (summarized in Figure 1) designed to quantify memorization, and assess whether the memorized content poses privacy risks.

### 3.1 EHR-FM Testing Framework

As most EHR-FM give black-box and prompt-only access to users, we designed our tests to evaluate whether prompts alone can trigger the disclosure of sensitive patient information. We first create four tests (T1-T4) designed to measure memorization under two domains of information extraction: Generative memorization and Embedding memorization (Figure 1). Generative memorization tests assess a model's ability to recreate data from the training cohort based on the following definition:

**Definition 3.1 (Generative memorization)** *Given a prompt $p$, consisting of a series of codes, a model $f$ can generate a time series of EHR records $s$ utilizing greedy decoding, where the concatenation of $[p||s]$ is part of a sample $x$.*

Embedding memorization tests focus on encoder-based EHR-FMs [32], and measure how much a model can memorize and reveal about a training sample through the embeddings as formalized below:

**Definition 3.2 (Embedding memorization)** *Given a prompt $p$, consisting of a series of codes, a model $f$ can generate an embedding $z$ that can be utilized to recover information associated with a sample $x$ in the training dataset.*

Finally, we argue that the degree of memorization does not always align with the degree of risk to patient privacy. Therefore, 2 additional tests (T5-T6) are presented to assess individual-level risk and to distinguish between patient-level memorization and population-level generalization.

## 3.2 Testing Setup

All proposed tests measure memorization as a function of: (1) the amount of information an adversary uses to reveal information, and (2) the risk associated with the revealed information. Throughout the tests, we use the following setups to reflect the amount of information an attacker has access to:

- **Random**: No information is provided to the models and it generates embeddings or sequences only from its prior. This reflects the internalized knowledge of a model.
- **Static**: The model is prompted with any demographic attributes it has been trained on, such as age or biological sex. These represent easily accessible and public information about individuals.
- **N-codes**: The model is prompted with the first N codes of a patient record. Such prompts require more information about an individual patient.

We can also curate more sophisticated categories of prompts. For instance, prompting the model with information about an individual's medication history. Different prompt setups will assess memorization based on what information can be accessed by an adversary and would give insight into what information has a higher likelihood of revealing private information.

To reflect the risk associated with revealed information, we select a number of codes, such as ICD-10 medical codes, related to diagnoses with social stigma as representatives of high-risk sensitive attributes. This information is classified under infectious diseases, substance abuse, and mental health conditions (see Table 4 in the appendix). We have selected these categories, as they are highly protected under regulations such as HIPAA and GDPR [1], often involving conditions that are poorly understood and associated with social taboos. The choice of sensitive attributes can be different depending on the setting or datasets, and we provide this categorization as an example.

## 3.3 Benchmark Implementation

We demonstrate our testing framework on EHRMamba2 [14]. We select this model as a representative benchmark because it is one of the few existing models designed for both patient embedding learning and EHR generation (forecasting). Additionally, its publicly available architecture details enable reproducible training. Also, EHRMamba2 is trained on the public MIMIC-IV dataset [23], which enables direct testing of memorization on known training samples, unlike other released models that are trained on private and inaccessible datasets.

# 4 Objective I: Measuring Memorization in EHR-FM

We evaluate EHR-FM *generative memorization* and *embedding memorization* with tests designed to measure the amount of training information extractable from the model and the risk associated with the information.

## 4.1 T1 - Trajectory memorization test

T1 examines how similar a sequence of generated codes is to a patient trajectory from the training data. Quantifying EHR sequence similarity is a nontrivial task, with two key nuances: (1) EHR code similarity may depend on their clinical context, for instance, two medications treating the same condition should have a higher similarity score compared to unrelated codes. (2) EHR data is inherently time-series-based; therefore, identical codes appearing at different time points should be penalized when measuring similarity between sequences.

We develop a similarity metric to quantify EHR sequence similarity. We first employ MedBERT [43], which adapts the BERT framework for EHR data, to map EHR codes to embeddings $h(\cdot)$ that capture clinical semantics (Figure 7 in the Appendix). This allows our distance measure to incorporate clinically meaningful semantics through the embeddings. Our distance score $d_{EMD}$ (Equation 1) then uses time-weighted Wasserstein or Earth Mover's Distance (EMD) [46] to measure the distance

between 2 sequences of embeddings, where $d(h(s1_i), h(s2_j))$ indicates the pairwise cosine distance between codes generated by MedBERT.

We incorporate the time difference between event tokens as a penalty for temporal alignment. The weight $T_{i,j}$ measures the visit time difference between token $i$ in one sequence and token $j$ in the other. EHR-FMs encode temporal information with time tokens, indicating the gap between each recorded code. The time penalty is determined based on how granular the EHR-FM record time is and how sensitive we want our score to be to time alignment. In our benchmark model, only time gaps larger than 1 hour are recorded, and we penalize every hour of misalignment with a weight of 1.

$$d_{EMD} = \inf_{T \in \mathcal{T}} \sum_{i=1}^{|s1|} \sum_{j=1}^{|s2|} T_{ij} d(h(s1_i), h(s2_j)), \tag{1}$$

We validate the functionality and sensitivity of our proposed metric in Appendix A using synthetic patient trajectories, analyzing how the measured distance changes across variations in patient trajectories, related and unrelated medications, and diagnoses.

**Analysis:** We compare the average distance (measured by $d_{EMD}$) of generated codes to the true trajectories for different prompt setups (Random, Static, 10, 20, and 50 codes), on our benchmark model. Figure 2a shows the distance over 100 prediction codes ($|s| = 100$) for 3K individuals in the pretraining cohort. Following the strategy of language models [9], for every prompt, hundreds of trajectories are sampled, and the distribution is used to quantify memorization.

We observe that **prompting the model with more information about an individual results in better sequence predictions** (lower $d_{EMD}$). This shows that the more information an attacker provides about an individual, the likelier it is for the model to recreate a patient EHR trajectory. However, distance measures are relative and may not easily translate to risk. Samples performing worse than random indicate that the prediction is no better than a random guess by the model. The high variance around the values indicates that forecast performance varies significantly between individuals, which highlights the importance of individual-level investigation on best-performing samples.

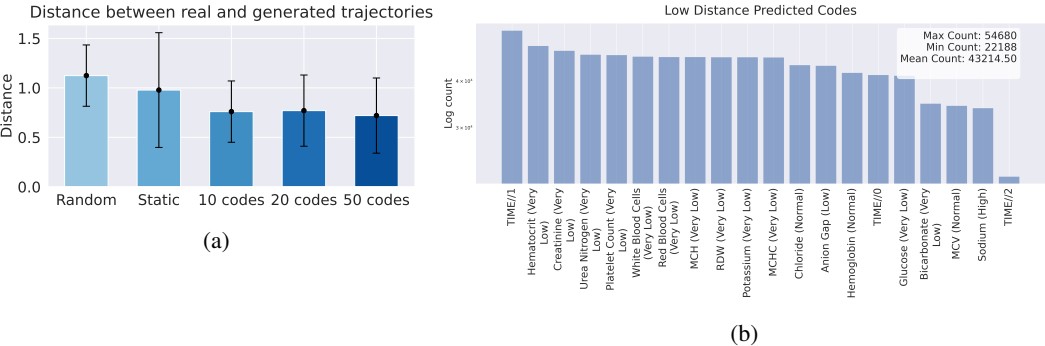

Figure 2: Model predictions in trajectory memorization. (a) Average distance between generated and ground truth EHR sequences of 100 codes for different prompting setups. (b) Predicted tokens and their frequencies in the best-performing trajectory prediction. All tokens are related to common lab measures, and most often normal values.

While revealing any information about an individual's health record is undesirable, the risk associated with the memorized information can significantly vary. For example, noting a low white blood cell (WBC) count poses far less risk than revealing highly sensitive details, such as a patient's HIV status. Our patient-level investigation of high-performing samples reveals that **trajectories with a smaller distance do not necessarily imply a higher risk**. In fact, these predictions correspond to short trajectories with common codes such as routine lab measures with normal values.

Figure 2b shows the top predicted codes for the best-performing prompts, prominently featuring common lab codes and time-related tokens. Our results further show that the model closely approximates the true token frequency distribution (Figure 3a). However, it under-generates less frequent codes, as

evidenced by the deviation from the diagonal in the low-frequency range. This shift indicates that the model favors more frequent codes, revealing a bias against generating rare codes.

Measuring similarity offers an intuitive and quantitative signal for memorization, but it does not directly capture risk. To address this, we follow up T1 with T2, which explicitly targets codes whose leakage could pose harm to patients—thus quantifying memorization risk more concretely.

## 4.2 T2 - Sensitivity test

Leaked information poses different risks, depending on the sensitivity of that information. The objective of T2 is to evaluate the likelihood of a model memorizing and revealing high-risk information about an individual. This information is represented as sensitive attributes defined in Section 3.2. We adopt a similar setup to T1 but explicitly remove any codes directly related to the sensitive condition from the prompt. T2 then measures the probability that the model reveals the sensitive attribute in its generated trajectory. The test passes if the model does not leak the sensitive attribute. Conversely, if the generated sequence contains the attribute, it indicates a memorization event that must be further validated to assess the associated privacy risk.

Table 1: Performance of the EHR-FM benchmark in predicting sensitive diagnoses. Results are reported for three categories of sensitive attributes (infectious disease, substance abuse, and mental health) under different prompt conditions. Higher prompt lengths increase the likelihood of revealing sensitive codes, highlighting privacy risks associated with memorization.

| Sensitive attribute | Patient prevalence | Prompt | AUROC | AUPRC | Precision | Recall | Positive prediction count |
|---|---|---|---|---|---|---|---|
| Infectious disease | 0.0476 | Statics | 0.548 | 0.053 | 0.000 | 0.000 | 0 |
| | 0.0476 | 10 codes | 0.672 | 0.153 | 0.353 | 0.004 | 6 |
| | 0.0478 | 20 codes | 0.697 | 0.177 | 0.444 | 0.006 | 8 |
| | 0.0473 | 50 codes | 0.742 | 0.219 | 0.411 | 0.018 | 23 |
| Substance abuse | 0.0672 | Statics | 0.622 | 0.090 | 0.000 | 0.000 | 0 |
| | 0.0665 | 10 codes | 0.706 | 0.171 | 0.467 | 0.023 | 49 |
| | 0.0661 | 20 codes | 0.719 | 0.191 | 0.594 | 0.024 | 41 |
| | 0.0665 | 50 codes | 0.751 | 0.267 | 0.577 | 0.084 | 162 |
| Mental health | 0.0611 | Statics | 0.604 | 0.081 | 0.000 | 0.000 | 0 |
| | 0.0605 | 10 codes | 0.669 | 0.138 | 0.889 | 0.004 | 8 |
| | 0.0608 | 20 codes | 0.681 | 0.170 | 0.733 | 0.007 | 11 |
| | 0.0604 | 50 codes | 0.724 | 0.286 | 0.721 | 0.114 | 202 |

**Analysis:** We evaluate T2 on the benchmark model for different levels of prompting, and the findings are summarized in Table 1. With short prompts (Static baseline) the predictive performance of the model is near random. Showing that the benchmark doesn't reveal sensitive information about a patient, only given the individual's age or demographics. However, the model's ability to reveal sensitive attributes increases with an increasing amount of prompt information. For an individual-level analysis, we select a conservative threshold of 30% and select all positive predictive prompts that generate the sensitive attribute in more than 30% of their trajectories (right column of Table 1). These identify potential problematic samples, but the risk of memorization depends on the context provided to the model. In fact, **not all positively predicted samples reflect memorization**. A well-performing model, like a skilled practitioner, should make informed predictions based on meaningful prompts, not mere memorization.

A sample fails the sensitivity test when the model generates a sensitive attribute even though it was excluded from the prompt. These are the most problematic cases where the model predicts a condition based on irrelevant input, indicating that it may have memorized spurious correlations or, more concerningly, details about a specific individual. For instance, one of our positive samples included a lab result showing a very low Absolute CD8 Count, the model predicted an HIV diagnosis in over 30% of sampled trajectories. While this does not necessarily indicate memorization, it reflects meaningful reasoning. Low CD8 counts are commonly observed in HIV patients and, although not sufficient for a full diagnosis, they increase the likelihood of HIV being suspected.

In contrast, we identify prompts that lack clear indicators of disease, yet the model still predicts a sensitive diagnosis. For example, one of our prompts was a 48-year-old individual transferred to the ED with a history of falls and discharged shortly after, representing minimal clinical detail. The model predicts alcohol abuse in 31% of the estimated trajectories. Despite the absence of relevant cues, the model reveals a sensitive diagnosis, causing this sample to fail the sensitivity test. We flag these prompts for further investigation in future test T5 to further assess their risk to patient privacy.

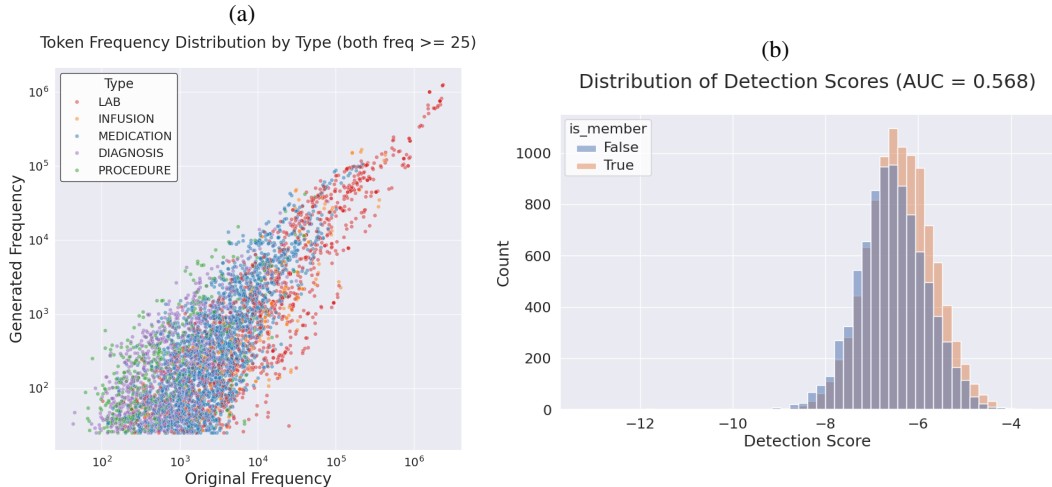

Figure 3: Code generation and membership inference. (a) Frequency of EHR code randomly generated by EHRMamba2 versus the original distribution of codes in the training dataset. (b) Distribution of detection scores from a membership inference attack based on model confidence. The x-axis shows scores, the y-axis shows counts, with colors indicating member (True) vs non-member (False) samples.

### 4.3 T3 - Probing test

T3 employs a probing test [2, 8] to evaluate memorization by examining information stored in embeddings. We train a probing model $g(\cdot)$ to predict sensitive information from embeddings, using either external data or a subset of the training dataset. This setup reflects the information available to an adversary, and T3 measures memorization risk as a function of how much knowledge the probe requires to recover the sensitive content. If probes successfully extract sensitive information from embeddings, developers should apply additional post-processing to remove such information before releasing the model. We validate our proposed T3 test on a synthetic temporal dataset (as shown in Appendix B) to demonstrate it successfully identifies memorized samples in a controlled setting with positive control cases.

**Analysis:** We apply the probing test to our benchmark model by prompting it with the first $N$ codes from a patient record. Embeddings are extracted up to the first 10, 20, or 50 tokens and remain frozen during probing. We then train a classifier on the embeddings to predict sensitive diagnoses ($\hat{y} = g(f(s))$), as defined in Section 3.2, where $f(s)$ denotes the foundation model embeddings and $\hat{y}$ the predicted diagnosis. If the classifier successfully infers the label from the embedding, we consider the model to have memorized sensitive information. The highest-risk scenario arises when the probing model, trained on a separate test cohort, can still successfully extract sensitive codes. This is particularly concerning, as it suggests that an adversary equipped with an external dataset and the trained model could recover sensitive information about individuals in the training cohort.

We train the probing model with embeddings of a separate test cohort as well as varying fractions of training data. High accuracy with minimal data suggests strong memorization rather than generalization. Table 6 in the appendix reports probing attack performance on sensitive diagnoses, measured across different prompt lengths (10, 20, 50 tokens) and training. **Extracting sensitive information only from the embeddings is difficult**, even if an adversary can access a portion of the training data.

AUROC values across all sensitive attributes remain around 0.5, indicating no clear memorization signal and suggesting random performance. Notably, AUPRC and F1 generally decline with increased training data, hinting at inconsistent memorization patterns. These trends may be partially influenced by dataset size variations: 113,579 for the test set, 102,222 for 0.1%, and 90,864 for 20%. If probing performance does not improve even with access to training data, as we see with our benchmark, then the model passes T3. Although, the sensitive signal may be encoded only within a specific subgroup, prompting T6, which targets subgroup-level probing to uncover such hidden memorization.

## 4.4 T4 - Membership inference

T4 evaluates Membership Inference in EHR-FM; specifically, an adversarial attack that reveals whether a specific sample was included in a model's training cohort [12]. Many MI attacks succeed using only model outputs such as logits [48, 13, 33], underscoring that risk extends even to EHR-FMs released as black-box models. This type of risk differs from direct patient data exposure as it requires an attacker to possess prior knowledge about an individual and then confirm their presence in the training data. When such inference succeeds, it can reveal sensitive details about the individual, including the institution or location of their clinical care or other training cohort–related information.

**Analysis:** We prompt the model with part of the patient EHR trajectory and extract the estimated logits for each token. Following the strategy introduced in [47], we estimate the membership score as the expected value of the log probabilities of the least probable tokens. The detection scores are shown in Figure 3b for 10K samples from the train and test cohort. The slight distribution separation suggests a marginally higher detection score for member samples, but the difference is not statistically significant, indicating weak discriminative power, suggesting the benchmark model does not substantially reveal membership status. These results are based on embeddings extracted from 1024 tokens of an individual trajectory, yet no clear separation emerges. Importantly, membership inference would become far more concerning if successful with substantially less information about the individual.

To meaningfully assess patient privacy risk, we should also evaluate MI at the individual level, focusing on which specific pieces of information trigger the model to reveal membership. Simply detecting membership does not alone indicate a privacy risk, especially in models trained on de-identified data. In order to pass T4, any prompt that results in a successful membership inference should not uniquely identify an individual. If it does, for example the prompt includes a unique diagnosis, this puts the individual in significant privacy risk.

## 5 Objective II: Evaluating Risk of Memorization

Most efforts to quantify the privacy implications of large foundation models focus on measuring information leakage [37]. While leakage is undesirable, our tests show that it doesn't always translate directly into patient privacy risk in healthcare. In this setting, leaking a single sensitive code about a patient can be much more concerning than exposing an entire blood panel. Hence, in our evaluation of memorization in EHR-FMs after going through T1-T4 to quantify memorization and assess the information leakage, we introduce two tests to investigate memorization privacy risk through: (1) Understanding whether the information leakage was the result of the model memorizing patient-level information or the model learning generalizable knowledge. (2) Evaluating sensitivity of different subgroups to memorization where information leakage can pose significantly different privacy risk.

## 5.1 T5 - Perturbation test

T5 evaluates whether adversarial prompts (such as the ones that identified sensitive attributes in T2) trigger patient-specific memorization or broader statistical patterns. **A sample is memorized and not generalized if the model output is sensitive to personal identifiers.** We test for this by constructing a set of perturbed prompts, where we modify personal identifiers such as age, ethnicity, or rare diagnosis codes, while keeping the rest of the prompt unchanged. We deliberately select these identifiers to reflect attributes unique to the target individual. We then re-evaluate the model's behavior on the perturbed prompts to determine whether the information leakage stems from memorization of a specific training individual or from generalizable patterns. If the model stops producing sensitive information after altering the identifiers, it signals that the original output resulted from memorizing

an individual instance. These cases indicate high-risk memorization and call for targeted safeguards to protect patient-level privacy.

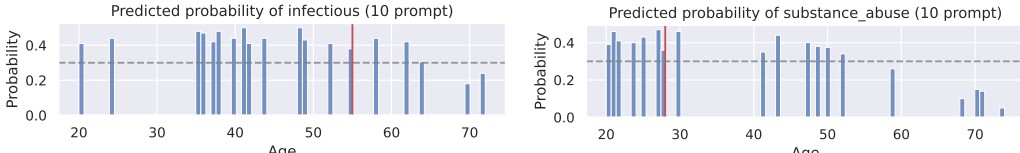

Figure 4: Predictive results for the perturbation test on 2 identified samples from infectious disease and substance abuse. Each bar represents the probability of generating the sensitive token, when the model is prompted with the 10 prompts and with different ages. The horizontal line is the threshold of 0.3 used in T2, and the red vertical line represents the original patient age.

**Analysis:** Our benchmark model uses only age as a demographic input. For prompts that revealed sensitive attributes in T2, we perturbed the age to test whether the model still predicts the sensitive diagnosis for people of different ages. Figure 4 shows two cases where the model correctly predicts Hepatitis C (left) and Alcohol abuse (right) for an individual whose age is marked by the vertical red line. Consistent predictions across individuals with similar ages suggest that the model relies on general trends for making the prediction rather than memorizing individual-specific patterns. However, we observe different outcomes when personal characteristics diverge. For example, in the case of substance abuse, predicted likelihoods drop for older individuals, revealing age-dependent patterns the model has learned. T5 is triggered when small changes to personal identifiers, such as age, cause significant shifts in prediction likelihoods. This sensitivity indicates that the model has memorized individual-specific information rather than generalizable patterns, posing a high risk to patient privacy. More examples of the perturbation test are shown in Figure 8 in the Appendix.

## 5.2 T6 - Sub-population test

In healthcare, we must assess memorization not only at the individual level but also across subgroups. Certain demographic or clinical populations face elevated privacy risks due to the sensitivity or rarity of their data. With our subgroup memorization test, we evaluate whether **belonging to a specific subpopulation increases the risk of memorization.** For example, if an attacker prompts the model with a rare condition code and the model leaks information about individuals with that condition, it exposes a serious privacy threat, one that targets small identifiable groups.

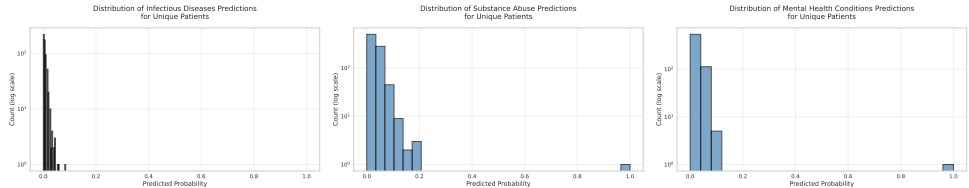

Figure 5: Likelihood of predicting the codes related to sensitive attributes (Infectious disease, mental health, and substance abuse) in EHR trajectories generated by prompting the benchmark model with rare diagnoses.

**Analysis:** We analyzed patients with rare diagnoses or procedures that appear only once in the training data. These individuals face a higher risk of re-identification due to the uniqueness of their records, as highlighted in genomic privacy research [18]. To test for memorization, we prompted the model with the patient's age and a rare diagnosis or procedure code, then applied the sensitivity test from T2 to detect any disclosure of sensitive attributes. Figure 5 shows the model's likelihood scores. Low scores suggest the model does not expose private information based on rarity alone, while high scores signal potential privacy concerns that require closer examination.

Our benchmark model revealed two concerning cases: (1) Sedative, hypnotic, or anxiolytic dependence, and (2) Mild manic episode without psychotic symptoms. These codes strongly indicate the sensitive attribute under test. Even if the model didn't directly memorize them, their presence in

high-likelihood outputs raises serious privacy concerns. If statistical associations cannot explain these predictions, they reflect memorization failures.

We also examine the elderly subgroup—patients over 85—who comprise about 5% of the MIMIC dataset. As a long-tailed cohort, they are more prone to overfitting and require targeted evaluation. We apply tests T1–T5 to assess memorization relative to the general population (Table 5, Appendix). While the ideal outcome remains no memorization, this subgroup analysis is critical: if a memorized instance does occur, the privacy risk is amplified because these patients belong to a more easily identifiable cohort. Any flagged cases therefore warrant heightened caution, since individuals in long-tailed groups are often **more uniquely identifiable**, even in de-identified datasets, underscoring the **need for stronger privacy safeguards**.

## 6 Discussion

We propose a new perspective on evaluating EHR foundation model memorization, grounded in the practical realities of healthcare. As EHR-FMs integrate into clinical workflows for documentation, decision support, and analytics [22], they often operate on sensitive subsets of patient records that carry high clinical relevance. If these models leak memorized patient information, this compromises privacy, misleading clinical decisions [4], and enabling adversarial attacks, posing serious ethical and legal challenges.

High-profile breaches, such as Stanford Hospital's 20M lawsuit over leaked records [21], underscore the urgency of systematically auditing AI models in healthcare. To address this, we introduce an evaluation framework for EHR-FM that measures memorization and its associated privacy risks. Our tests quantify different forms of memorization and assess their implications in clinical settings, distinguishing harmful leakage at the patient level from useful generalization at the population level.

EHR-FMs continue to evolve, and as their capabilities grow, prioritizing privacy becomes critical. Our work equips developers with tools to detect memorization and sets the foundations for future mitigation strategies. Developers can use these tests to identify and address flagged samples, reduce memorization during training, and better understand risks, thresholds, and attacker strategies. Although our tests are not exhaustive, they provide a practical starting point. The open-source code enables others to adapt, extend, and build toward more secure EHR foundation models. Follow up study will investigate such risks across broader model categories and stakeholder perspectives.

**Acknowledgments and Disclosure of Funding**

ST was supported by the Eric and Wendy Schmidt Center at the Broad Institute of MIT and Harvard. LS was supported by the Wallenberg AI, Autonomous Systems and Software Program (WASP) funded by the Knut and Alice Wallenberg Foundation. MG was supported by the National Science Foundation (NSF) 22-586 Faculty Early Career Development Award (#2339381), a Gordon & Betty Moore Foundation award, a Google Research Scholar award and the AI2050 Program at Schmidt Sciences. Resources used in preparing this research were provided, in part, by the Province of Ontario, the Government of Canada through CIFAR, and companies sponsoring the Vector Institute.

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

# A  Validation of T1 similarity metric

In this paper, we present a similarity metric to measure the similarity between two sequences of structured EHR data, including diagnoses, medication records, lab measures, etc. All this information is stored as a sequence of codes (tokens), with special time tokens in between that indicate the time gap between the recorded codes. Our similarity metric needs to measure the degree of similarity between codes, and the similarity in the sequence of events. Large language models trained on medical data successfully encode medical text into representations that preserve clinically meaningful semantics [43]. We can leverage these models to quantify the similarity between EHR codes. This is critical in measuring the similarity or distance between 2 sequences of EHR records because the similarity between these codes can be dramatically different. For instance, the similarity between similar antibiotics should be higher than a pain medication and an antibiotic. Next, our metric should assess how 2 sequences have a similar timeline, and for that, we use the time codes to penalize codes that are placed in different time windows. This penalty increases as the codes are further apart in the sequence. Note, that we can only penalize time depending on how granular timelines are processed in the EHR-FM. Our Benchmark, for instance, processes time with a granularity of one hour. This means all codes collected within an hour will be considered to have the same time.

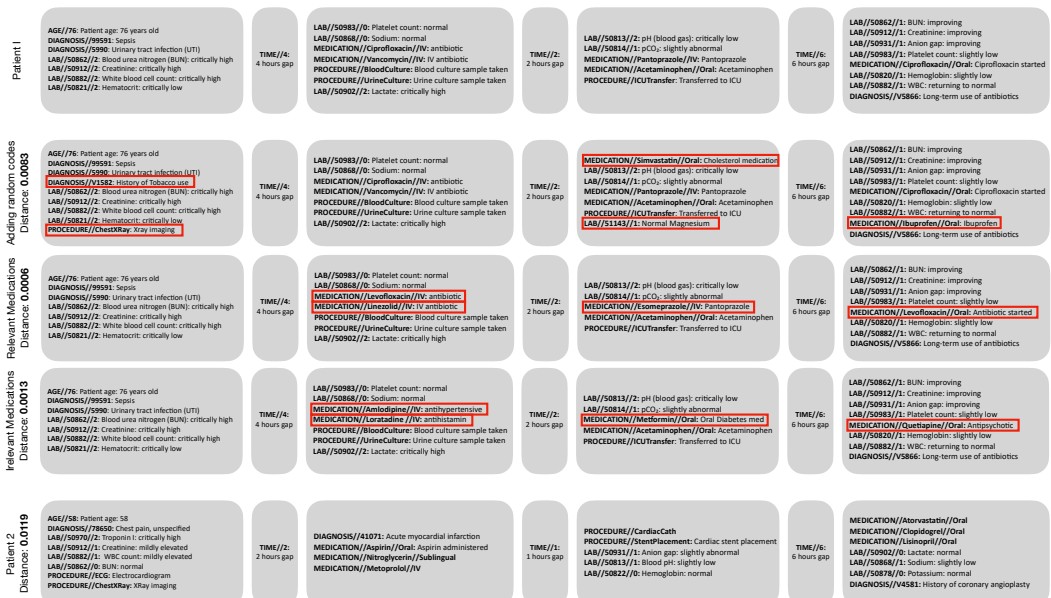

Figure 6: An example of different synthetic patient records and how the proposed metric measures the distance between them. The first row represents our reference sample Patient 1. Each subsequent row represents a different trajectory, with the differences highlighted.

We demonstrate the functionality of our score using the EHR record of a hypothetical patient. A 76-year-old individual was admitted to the ICU with a Urinary tract Infection (UTI) and eventually developed Sepsis. The first row of Figure 6 demonstrates the EHR record for this individual.

To show the sensitivity of our metric, we measure the distance of this sequence to several sequences, created with careful perturbations as described below:

1. Randomly adding or removing codes to the EHR sequence to test sensitivity to sequence length.

2. Replacement of medications with clinically equivalent alternatives (for example, swapping antibiotics) or irrelevant medications to assess if the metric captures the different level of differences.

3. Modifying lab results to simulate errors, such as changing infection indicators from high to low.

4. Compare distances between completely different patients to assess the score for unrelated sequences.

The red boxes highlight the tokens that are perturbed with respect to the reference patient. As shown in Figure 6, our metric effectively distinguishes clinically relevant changes in patient trajectories. Replacing medications with similar alternatives results in smaller distances compared to using irrelevant medications. Altering lab values leads to greater distances due to their density and diagnostic relevance, while distances between unrelated patients are higher.

# B  Demonstrating Detection of Memorization with known ground truth (positive control experiment)

Validating positive controls in real-world EHR data is inherently challenging due to the absence of ground truth regarding which samples may have been memorized by a model. To address this, we designed a controlled synthetic experiment that mimics the structure of EHR-like sequences and allows us to explicitly simulate memorization behavior. In this setup, each sequence consists of digits (analogous to diagnosis codes), where higher digits occur less frequently, following a skewed distribution $p = \frac{1}{(d+1)^2}$, reflecting the prevalence imbalance commonly observed in clinical data. We implemented a toy foundation model that learns embeddings from these sequences and generates future sequences based on them. We added further stochasticity to the generative process to simulate more complex setups like EHR-FM.

To simulate memorization, we encoded a deterministic rule in the model: if a sequence begins with the digits [0,1], a specific embedding dimension is set to 1; otherwise, it remains 0. The remaining dimensions are learned normally. In generation, this memorized embedding dimension directly affects the output; If set to 1, the model is forced to produce a rare digit (e.g., 9), emulating the behavior of memorizing rare or sensitive clinical cases.

| Percentage of Data Used | Accuracy | AUROC | AUPRC | F1 Score | Precision | Recall |
|---|---|---|---|---|---|---|
| 20% | 0.931 | 0.934 | 0.906 | 0.924 | 0.995 | 0.862 |
| 10% | 0.917 | 0.919 | 0.883 | 0.900 | 0.975 | 0.836 |
| 5% | 0.875 | 0.916 | 0.860 | 0.833 | 0.922 | 0.759 |
| 1% | 0.742 | 0.876 | 0.665 | 0.602 | 0.728 | 0.512 |

Table 2: T3 results on simulated temporal data for different size of training data.

We then applied our probing tests to this synthetic model to assess whether T3 could successfully detect memorized samples. Table 2 presents the results, demonstrating how effectively an adversary can infer memorized content depending on the amount of information they have access to. hen the adversary has access to limited data for training (1% setting), only 74% of cases are identified and leaked. When the probing model is trained on a larger fraction of the input, its ability to recover memorized outputs improves substantially, reaching a precision of 0.92. This confirms that our test reliably detects memorization under varying levels of adversarial knowledge.

We also validate the probing test in our controlled synthetic setup using explicit positive controls. Specifically, we take a memorized sequence, defined as one that begins with [0, 1], which deterministically produces the rare digit 9 in the generated output due to its hard-coded influence in the embedding (Table 3). To test the sensitivity of our perturbation framework, we generate 9 additional sequences by modifying only the first digit (i.e., replacing the 0 with each digit from 1 to 9), while keeping the rest of the sequence unchanged. For each perturbed input, we generate 1000 trajectories. The perturbation test T5 then evaluates which fraction of the generated sequences still includes the rare digit 9. In the results of T5, the sharp drop in the fraction of sequences containing the sensitive code, as compared to the original [0, 1] input, confirms that the model has memorized a specific pattern, and the test is capable of revealing this behavior reliably, highlighting the harmful memorization behaviour of the model.

| Starts With | % of Sequences Containing 9 |
|---|---|
| 01... | 100.00% |
| 11... | 2.54% |
| 21... | 2.42% |
| 31... | 2.32% |
| 41... | 2.41% |
| 51... | 2.56% |
| 61... | 2.42% |
| 71... | 2.43% |
| 81... | 2.47% |
| 91... | 2.40% |

Table 3: Perturbation test (T5) results on the simulated temporal data. First row shows a reference samples, and the rest of the rows demonstrate perturbations to that sample. The sharp drop in the percentage of sequences generating a rare code (9) indicates that T5 is identifying a memorization behaviour.

## C   Categories of sensitive medical conditions

In the main text, we highlight a subset of attributes that are particularly sensitive due to their association with stigma, regulatory protection, and social taboos. Table 4 lists the specific ICD-10 codes we use as representatives of such high-risk categories, covering infectious diseases, substance abuse, and mental health conditions. While the selection is not exhaustive and may vary across datasets or contexts, it provides a concrete example of how sensitive information can be categorized for analysis.

| Category | Detail |
|---|---|
| Infectious Diseases | HIV/AIDS, Tuberculosis, Hepatitis B or C, Chlamydia |
| Substance Abuse | Dependence or abuse of alcohol, opioids, cocaine, heroin, cannabis, hallucinogens, or stimulants |
| Mental Health | Schizophrenia, Bipolar Disorder, Personality Disorders, Paranoia, Brief Psychotic Disorder, Post-Traumatic Stress Disorder, Anorexia Nervosa, Manic Episodes, Borderline Personality Disorder |

Table 4: Categories of high-risk medical conditions considered sensitive in model inference. Their disclosure—intentional or not—through machine learning models may pose ethical, legal, or privacy risks for patients.

## D   Generative memorization in foundation models

In the main text, we discuss subgroup memorization risks, with particular focus on rare conditions, procedures, and elderly patients. Table 5 presents the detailed results of the sensitivity test (T2) for the elderly subgroup. While no memorized instances were detected, the table illustrates the evaluation procedure and highlights why even small, long-tailed cohorts require close monitoring for potential privacy risks.

| Sensitive attribute | Patient prevalence | Prompt | AUROC | AUPRC | Precision | Recall | Positive prediction count |
|---|---|---|---|---|---|---|---|
| Infectious disease | 0.0328 | Statics | 0.500 | 0.033 | 0.000 | 0.000 | 0 |
| | 0.0476 | 10 codes | 0.517 | 0.048 | 0.000 | 0.000 | 0 |
| | 0.0112 | 20 codes | 0.51 | 0.048 | 0.000 | 0.000 | 0 |
| | 0.0120 | 50 codes | 0.49 | 0.012 | 0.000 | 0.000 | 0 |
| Substance abuse | 0.0574 | Statics | 0.500 | 0.057 | 0.000 | 0.000 | 0 |
| | 0.0062 | 10 codes | 0.516 | 0.007 | 0.000 | 0.000 | 0 |
| | 0.0052 | 20 codes | 0.515 | 0.006 | 0.000 | 0.000 | 0 |
| | 0.0055 | 50 codes | 0.465 | 0.006 | 0.000 | 0.000 | 0 |
| Mental health | 0.0170 | Statics | 0.500 | 0.017 | 0.000 | 0.000 | 0 |
| | 0.0162 | 10 codes | 0.524 | 0.018 | 0.000 | 0.000 | 0 |
| | 0.0163 | 20 codes | 0.566 | 0.027 | 0.000 | 0.000 | 0 |
| | 0.0153 | 50 codes | 0.565 | 0.055 | 0.000 | 0.000 | 0 |

Table 5: Performance comparison of model inference for sensitive medical conditions in older patients (age $\geq 85$) versus the general population, across different prompt strategies.

| Sensitive Attribute | Prompt | AUROC | | | | AUPRC | | | | F1 | | | |
|---|---|---|---|---|---|---|---|---|---|---|---|---|---|
| | | test | 0.1% | 10% | 20% | test | 0.1% | 10% | 20% | test | 0.1% | 10% | 20% |
| Infectious Diseases | 10 codes | 0.5470 | 0.534 | 0.540 | 0.544 | 0.0687 | 0.067 | 0.068 | 0.068 | 0.0687 | 0.342 | 0.138 | 0.147 |
| | 20 codes | 0.5547 | 0.540 | 0.540 | 0.544 | 0.0804 | 0.068 | 0.138 | 0.147 | 0.0804 | 0.273 | 0.092 | 0.105 |
| | 50 codes | 0.5661 | 0.544 | 0.544 | 0.544 | 0.0955 | 0.068 | 0.147 | 0.147 | 0.0955 | 0.248 | 0.105 | 0.105 |
| Substance Abuse | 10 codes | 0.5000 | 0.500 | 0.527 | 0.500 | 0.0668 | 0.067 | 0.077 | 0.067 | 0.0668 | 0.107 | 0.111 | 0.000 |
| | 20 codes | 0.5000 | 0.500 | 0.527 | 0.500 | 0.0668 | 0.067 | 0.111 | 0.000 | 0.0668 | 0.193 | 0.078 | 0.000 |
| | 50 codes | 0.5475 | 0.500 | 0.500 | 0.500 | 0.0890 | 0.067 | 0.000 | 0.000 | 0.0890 | 0.000 | 0.000 | 0.000 |
| Mental Health | 10 codes | 0.5000 | 0.537 | 0.500 | 0.500 | 0.0610 | 0.069 | 0.061 | 0.0611 | 0.0611 | 0.113 | 0.000 | 0.000 |
| | 20 codes | 0.5000 | 0.537 | 0.500 | 0.500 | 0.0611 | 0.069 | 0.000 | 0.000 | 0.0611 | 0.000 | 0.000 | 0.000 |
| | 50 codes | 0.5664 | 0.500 | 0.500 | 0.500 | 0.1060 | 0.061 | 0.000 | 0.000 | 0.1060 | 0.000 | 0.000 | 0.000 |

Table 6: Evaluation metrics for different conditions across code lengths and training fractions.

# E   Supplementary figures

Appendix Figure 7 provides an overview of the six memorization tests (T1–T6). It illustrates how we group the tests into generative, embedding-based, and privacy risk assessment categories, highlighting their respective goals: reconstructing training data, probing embedding leakage, and distinguishing memorization from generalization.

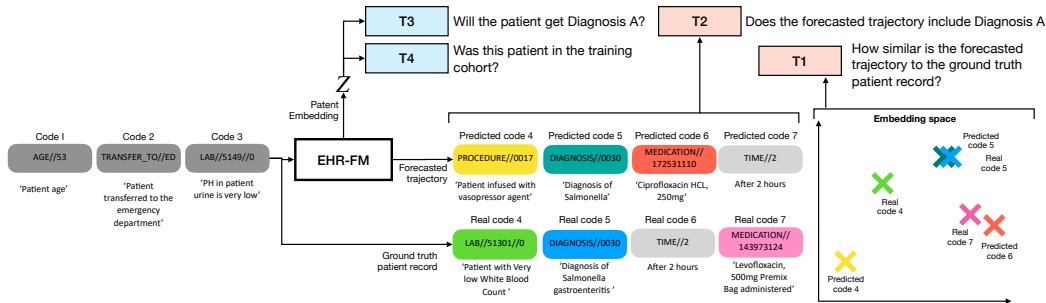

Figure 7: A hypothetical patient EHR trajectory fed into an EHR-FM model. The model generates a representation $Z$ as well as a forecast of the future codes in the EHR trajectory. T1 measures the similarity of the generated and ground truth sequence using a medical LLM. T2 measures the likelihood of the model generating sensitive tokens in its trajectory. T3 and T4 investigate if $Z$ is predictive of deficit health conditions or can determine membership of a sample in the pre-training.

T5 evaluates whether adversarial prompts trigger patient-specific memorization or broader statistical patterns. To further illustrate this analysis, we include additional examples of perturbed prompts

and corresponding model outputs. These results complement Figure 4 by showing how altering personal identifiers such as age affects predictions of sensitive diagnoses. Figures 8 and onward provide extended cases, highlighting instances where the model either continues to reveal sensitive information (suggesting memorization) or no longer does so (indicating generalization).

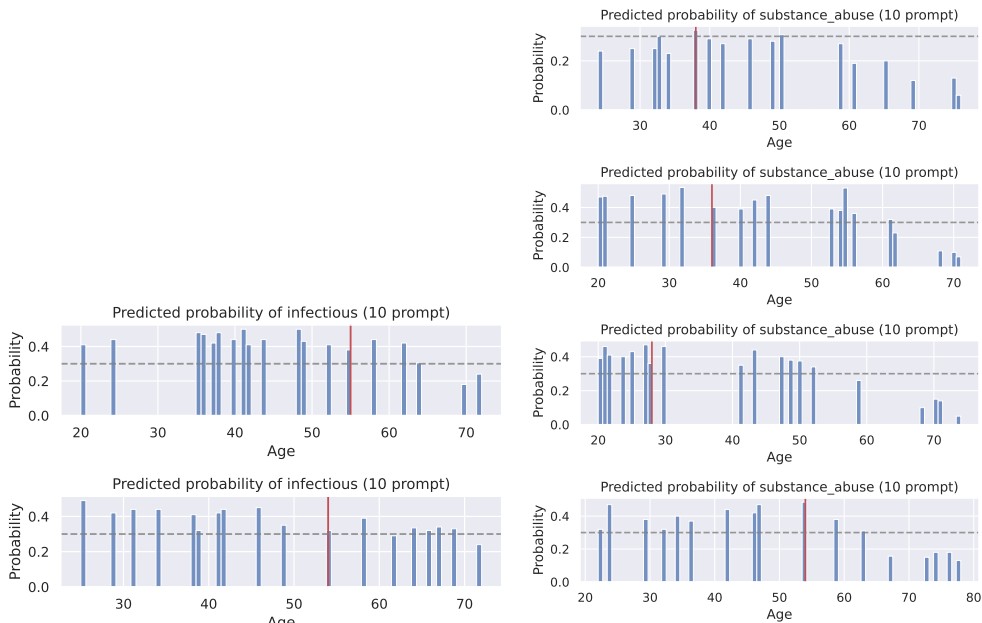

Figure 8: Examples of perturbed adversarial prompts for predicting Infectious disease (left column) and substance abuse (right column).

