# OpenReview forum: "An Investigation of Memorization Risk in Healthcare Foundation Models"
_NeurIPS.cc/2025/Conference — NeurIPS 2025 poster_

### Official Review · Reviewer_Reek · 2025-06-03

**Clarity:** 3
**Significance:** 1
**Originality:** 3
**Rating:** 3
**Confidence:** 4

**Summary:**

The paper presents a suite of tests to study memorization of foundational models in EHR prediction tasks. Some initial experiments are reported.

**Questions:**

Is there empirical evidence that the tests actually reflect differences between models that are controlled for memorization, sensitive attributes, or spurious correlations?

Are there analyses with a more substantial set of prompting and cases? The present tests seem to be limited.

**Ethical Concerns:**

["NO or VERY MINOR ethics concerns only"]

**Final Justification:**

Thank you for the detailed rebuttal and the additional experiments. I don't have any additional major concerns, and I appreciate the clarifications. I will raise my score, but I maintain a slightly negative position.

**Limitations:**

yes

**Paper Formatting Concerns:**

Some grammatical errors and simple formatting errors like double dots.

**Quality:**

2

**Strengths And Weaknesses:**

About the general problem setup:

The paper tackles an interesting problem of studying the performance of foundational medical coding models and their tendency to be biased under several measures.

The paper is concerned with the models that reveal sensitive information by memorization. I am not sure if I can fully agree with this concern. Generating a trajectory of treatments with a statement that the greater the input information value, the more likely the model is to create the patient trajectory or sensitive output. Isn’t this contradictory? Wouldn’t we want the model to provide accurate trajectories with precise background information? Isn’t it the whole point that medical treatments should follow the prognosis and medical history, and each patient should get the most fitted treatment trajectory — thus, if the information is exactly the same as with another patient, then the trajectory should be identical? Surely, with less information, the model will likely make predictions that are less precise.
For instance, the HIV example shows that the model predicts codes that are sensitive, but it does so with some likelihood and based on valid input information. What is the problem here? I think spurious correlations will be likely to exist in any model in some quantity, but the examples given did not convince me that the test suite would reveal something that we could not expect.

Results and their significance

In general, the manuscript does not communicate the severity of the problem, and the results are not discussed and compared across different models, methods to control for bias, and spurious correlation controls. Many of the results are unclear. For instance, T2: not clear what exactly are the results presented in Table 1 and how they have been computed. The same issue is with the tables in the Appendix.

The paper has a great initiative, but I think that the identification of the actual problems that may arise need to be sharpened and the empirical results need to be strengthened to include envidence of the tests capturing effects when the problems are controlled (e.g. test a model where sensitive controls are present in training, finetuning or post-processing to improve interpretability vs. a model where these are not present).

---

> ### Author Rebuttal · Authors · 2025-07-30
>
> We thank the reviewer for recognizing the importance of investigating the behavior of foundation models in medical coding. We have carefully addressed all suggestions to improve clarity and respond to the specific comments below.
>
> **Generalization vs. Memorization and Data Leakage Risks:**
> We thank the reviewer for raising this important point. We fully agree that clinical prediction models should leverage informative patient histories to generate accurate and personalized treatment trajectories. In fact, part of the main purpose of our work is to draw attention to the distinction between model generalization and harmful memorization that can leak sensitive information from training data, even when prompted with partial or non-specific inputs.
>
> The concern is not that models generate plausible sequences given rich clinical histories, which is expected. Rather, the risk arises when high-confidence predictions of sensitive conditions (e.g., HIV status, substance use) can be elicited from incomplete or generic inputs. This may signal that the model is not generalizing from population-level patterns but is memorizing individual cases seen during training. For instance, in our substance use example, a patient with repeated falls was assigned a sensitive diagnosis with no clear clinical justification in the input. This suggests the model may be reproducing memorized associations from similar training examples, raising serious privacy concerns.
>
> Our framework is explicitly designed to detect such privacy leakage through memorization. Tests T1–T4 focus on scenarios where an adversary, with limited but plausible access to background information (e.g., demographics, early labs), could prompt the model and recover sensitive attributes about a patient. Such inputs could realistically be inferred from public EHRs (e.g., MIMIC-III), leaked insurance claims, or personal disclosures on social media. For instance:
> T1, T2, and T6 demonstrate what an adversary might infer with access to partial knowledge (e.g., a few diagnosis codes) from an individual.
>
> T3 models the leakage potential when the adversary gains access to the embeddings or limited training samples.
>
> T4 estimates whether an adversary can determine if a given patient was part of the training cohort, based on access to a partial view of their medical records.
>
> To distinguish generalization from leakage, we apply perturbation-based tests: if the model still outputs rare, sensitive codes when key identifier information is removed or substituted, this behavior is less likely due to robust reasoning and more likely due to memorized correlations or training data artifacts. These patterns are not always evident without systematic probing, which is precisely what our framework enables. We will further clarify this distinction in the next version to ensure the motivation behind our framework is made explicit.
>
> **Empirical Evidence of Memorization in a Controlled Setting:** We demonstrate that our framework can detect memorization in a non-healthcare domain with known ground truth, serving as a positive control experiment.
> Thank you for this insightful suggestion. Validating positive controls in real-world EHR data is inherently challenging due to the absence of ground truth regarding which samples may have been memorized by a model. To address this, we designed a controlled synthetic experiment that mimics the structure of EHR-like sequences and allows us to explicitly simulate memorization behavior.
> In this setup, each sequence consists of digits (analogous to diagnosis codes), where higher digits occur less frequently, following a skewed distribution $p=\frac{1}{(d+1)^2}$reflecting the prevalence imbalance commonly observed in clinical data. We implemented a toy foundation model that learns embeddings from these sequences and generates future sequences based on them. We added further stochasticity to the generative process to simulate more complex setups like EHR-FM.
> To simulate memorization, we introduced a deterministic rule: if a sequence begins with the digits [0,1], a specific embedding dimension is set to 1; otherwise, it remains 0. The remaining dimensions are learned normally. In generation, this memorized embedding dimension directly affects the output—if set to 1, the model is forced to produce a rare digit (e.g., 9), emulating the behavior of memorizing rare or sensitive clinical cases.
> We then applied our probing tests to this synthetic model to assess whether they could successfully detect memorized samples. Below, we present the results of these tests, which demonstrate the ability of our framework to flag memorization under controlled, interpretable conditions.
>
> | Percentage of Data Used | Accuracy | AUROC | AUPRC | F1 Score | Precision | Recall |
> |-------------------------|----------|--------|--------|----------|-----------|--------|
> | 20% | 0.931 | 0.934 | 0.906 | 0.924 | 0.995 | 0.862 |
> | 10% | 0.917 | 0.919 | 0.883 | 0.900 | 0.975 | 0.836 |
> | 5% | 0.875 | 0.916 | 0.860 | 0.833 | 0.922 | 0.759 |
> | 1% | 0.742 | 0.876 | 0.665 | 0.602 | 0.728 | 0.512 |
>
>
> These results illustrate how effectively an adversary can infer memorized content depending on the amount of information they have access to. For instance, when the probing model is trained on a larger fraction of the input (e.g., >20% of the sequence), its ability to recover memorized outputs improves substantially, reaching a precision of 0.92. This confirms that our test reliably detects memorization under varying levels of adversarial knowledge.
> We also validate the probing test in our controlled synthetic setup using explicit positive controls. Specifically, we take a memorized sequence—defined as one that begins with [0, 1]—which deterministically produces the rare digit 9 in the generated output due to its hardcoded influence in the embedding. To test the sensitivity of our perturbation framework, we generate 9 additional sequences by modifying only the first digit (i.e., replacing the 0 with each digit from 1 to 9), while keeping the rest of the sequence unchanged. For each perturbed input, we generate 1000 trajectories.
> The perturbation test T5 then evaluates what fraction of the generated sequences still include the rare digit 9. In the results of T5, the sharp drop in the fraction of sequences containing the sensitive code, as compared to the original [0, 1] input, confirms that the model has memorized a specific pattern, and the test is capable of revealing this behavior reliably.
>
> | Starts With | % of Sequences Containing 9 |
> |-------------|------------------------------|
> | 01...       | 100.00%                      |
> | 11...       | 2.54%                        |
> | 21...       | 2.42%                        |
> | 31...       | 2.32%                        |
> | 41...       | 2.41%                        |
> | 51...       | 2.56%                        |
> | 61...       | 2.42%                        |
> | 71...       | 2.43%                        |
> | 81...       | 2.47%                        |
> | 91...       | 2.40%                        |
>
> We are happy to include an extended version of this study in the appendix to validate the performance of our tests on positive control samples.
>
> **Clarification on Tables:**  In T2 results we report the percentage of individuals for whom the model reveals a sensitive diagnosis code in the generated EHR trajectories. We define a "revealed" case as one in which more than 30% of the generated sequences contain the sensitive code, indicating a high likelihood of unintended disclosure. These results are presented across varying levels of prompt, simulating scenarios where an adversary has access to different amounts of information about a patient. This setup allows us to estimate the risk of sensitive information leakage as a function of adversarial knowledge. The tables in the appendix show the same test for different subgroups (for instance elderly population). This helps us understand if the risk varies across subpopulations.
>
> **Evaluation on different models:** We want to note that our primary goal was to develop and validate an effective evaluation methodology, and not focus on the benchmarking task. Currently, there are only a handful of models available and they may not pose significant memorization concerns, the rapid advancement of EHR-FMs suggests that these risks may soon become more pressing. Raising awareness of these issues now is essential and we believe this work provides a strong foundation for future studies to assess memorization risks across a broader range of models.
> Nevertheless, We have extended our analysis to include MEDS-torch, another EHR-FM benchmark [1]. Below shows a summary of the T1 results for predicting 20 steps ahead. The results follow similar trends to those of EHR-Mamba2, with only slight variation in “50 prompt” settings. We are happy to include full results in the appendix to illustrate the generality and applicability of our approach.
>
> [1] https://github.com/Oufattole/meds-torch
>
> -------------------------------------------------------------------------
>
> |	Prompt Type   |	MEDS-torch (20)	|
>
> | **Random**      	|	0.701 +- 0.1376	|
>
> | **Static**      		|	0.669 +- 0.149      	|
>
> | **10 prompts**      	|	0.65 +- 0.162      |
>
> | **20 prompts**      	|	0.65 +- 0.1830      	|
>
> | **50 prompts**      	|	0.677 +- 0.144    |
>
> -------------------------------------------------------------------------

---

> > ### Author Response · Authors · 2025-08-06
> > **Follow-up**
> >
> > Dear Reviewer,
> >
> > We’re writing to follow up, as we haven’t seen any additional comments since the rebuttal phase.
> > In our rebuttal, we carefully addressed all the concerns you raised, and made sure to clarify all the questions. We conducted additional evaluations to demonstrate the functionality of our tests and clarified all details regarding prompt design. If there are any remaining questions or concerns, we would truly appreciate the chance to clarify.

---

> > > ### Comment · Reviewer_Reek · 2025-08-07
> > >
> > > Thank you for the detailed rebuttal and the additional experiments. I don't have any additional major concerns, and I appreciate the clarifications. I will raise my score, but I maintain a slightly negative position.

---

### Official Review · Reviewer_Zqbh · 2025-06-20

**Clarity:** 3
**Significance:** 3
**Originality:** 3
**Rating:** 5
**Confidence:** 4

**Summary:**

This paper proposes a suite of tests to detect memorization in Foundation Models trained on structured EHR (Electronic Health Record) data. The goal of the authors is to quantify the risk of LLMs leaking private information.

The authors create 6 tests and validate them on EHRMamba, a prompt-based model trained on MIMIC-IV. They prompt the model to generate patient medical code sequences varying the amount of ground truth information given to the model in the input prompt. Four of the tests (T1, T2, T5 and T6) use directly the code sequences generated by the EHRMamba, while the other two (T3 and T4) use the embeddings generated by the LLM.

**Questions:**

# Questions/Suggestions

1. Add a positive-control benchmark for each test

This is the main concern of the paper. For this method to be generalizable and used out-of-the-box in any medical LLM, there has to be evidence that these metrics are actually doing something and are able to detect memorization. For the paper to be accepted, there has to be statistical evidence for every test discriminates between positive and negative examples. There might be many ways of proving this, but one could be to overfit a model so that it always leaks information.

In general, claims should be backed up with more statistical rigor.

2. Demonstrate generality

It would be ideal to run the experiments on at least another medical model. However, without fixing point 1, these experiments wouldn't have value.

3. Present the prompts used

These might be in the attached code, but I couldn't access it. Regardless, it would be useful to have them in the paper too, so that results are reproducible and can be better analyzed.

**Ethical Concerns:**

["NO or VERY MINOR ethics concerns only"]

**Final Justification:**

The authors have addressed the key concerns convincingly, and since the rest of the paper was already clearly and well written, this warrants a positive recommendation for acceptance.

**Limitations:**

Yes.

**Paper Formatting Concerns:**

None.

**Quality:**

3

**Strengths And Weaknesses:**

# Strengths

## 1. Critical problem addressed

The paper addresses the very relevant yet usually neglected problem of Large Language Models generating patient private information that they were trained on. A working toolkit to quantify and detect leaked information could become a standard diagnostic before releasing medical LLMs.

## 2. Clear structure

The problem of memorization in healthcare LLMs is introduced clearly and the overall paper is well structured and easy to follow.

## 3. Sound testing framework and setup

The definitions of generative and embedding memorization are sensible, and the proposed testing setup of varying the amount of private information shown to the model is a good starting point to test for leaked training data.

## 4. Original metric use

The use of Wasserstein distance to quantify the similarity between generated and real patient trajectories is interesting and original.

# Weaknesses

## 1. No positive-control validation and anecdotal evidence of tests working

There is no statistical evidence that the tests would actually detect cases of memorization.

### T1

T1 is the only test that is stress-tested, but even then, only one synthetic perturbation example is shown in Appendix A. We don’t know if the scores shown would generalize or are just incidental.

### T2

For T2, another single anecdotal example is presented in lines 211-213: “given a prompt about a 48-year-old individual transferred to the ED with a history of falls and discharged shortly after—representing minimal clinical detail—the model predicts alcohol abuse in 31% of the estimated trajectories.” How do we know that this is a real failure to begin with and not the model capturing underlying valid patterns in the data?

### T3

In T3, in lines 236-237, the authors state: “AUROC values across all sensitive attributes remain around 0.5, indicating no clear memorization signal and suggesting random performance.” However, no positive example is presented. How do we know that the metric will work on positive examples?

### T4

In T4, no positive example is shown. Additionally, Figure 3b is analyzed and the difference between both distributions in it is considered not statistically significant, but no statistical test is mentioned.

### T5

In T5, in lines 283-285, again the authors state that “If the model stops producing sensitive information after altering the identifiers, it signals that the original output resulted from memorizing an individual instance.” but no proof is provided, and only negative examples are shown.

### T6

In T6, two positive examples are presented in lines 314-318, but even the authors are unsure whether these are valid examples that show the metric working, as they state: “If statistical associations cannot explain these 318 predictions, they reflect memorization failures.”

## 2. Experiments on single model

Experiments have been conducted on one model + one dataset. Generality is untested.

## 3. No mitigation proposed

Framework stops at detection and no mitigation strategy is offered.

## 4. Code not available

The code shows as not available to me, but this might be a technical issue unrelated to the authors.

---

> ### Author Rebuttal · Authors · 2025-07-30
>
> We thank the reviewer for their insightful feedback and thoughtful comments.  The review included several constructive suggestions, such as validating our tests using controlled positive examples, which we have carefully incorporated and believe  have helped improved the paper.
>
> **Positive control experiment**
> Thank you for this insightful suggestion. Validating positive controls in real-world EHR data is inherently challenging due to the absence of ground truth regarding which samples may have been memorized by a model. To address this, we designed a controlled synthetic experiment that mimics the structure of EHR-like sequences and allows us to explicitly simulate memorization behavior.
> In this setup, each sequence consists of digits (analogous to diagnosis codes), where higher digits occur less frequently, following a skewed distribution $p=\frac{1}{(d+1)^2}$reflecting the prevalence imbalance commonly observed in clinical data. We implemented a toy foundation model that learns embeddings from these sequences and generates future sequences based on them. We added further stochasticity to the generative process to simulate more complex setups like EHR-FM.
> To simulate memorization, we introduced a deterministic rule: if a sequence begins with the digits [0,1], a specific embedding dimension is set to 1; otherwise, it remains 0. The remaining dimensions are learned normally. In generation, this memorized embedding dimension directly affects the output—if set to 1, the model is forced to produce a rare digit (e.g., 9), emulating the behavior of memorizing rare or sensitive clinical cases.
> We then applied our probing tests to this synthetic model.
>
> | Percentage of Data Used | Accuracy | AUROC | AUPRC | F1 Score | Precision | Recall |
> |-------------------------|----------|--------|--------|----------|-----------|--------|
> | 20 | 0.93 | 0.93 | 0.90 | 0.92 | 0.99 | 0.86 |
> | 10 | 0.91 | 0.92 | 0.88 | 0.90 | 0.97 | 0.83 |
> | 5 | 0.87 | 0.91 | 0.86 | 0.83 | 0.92 | 0.75 |
> | 1 | 0.74 | 0.87 | 0.66 | 0.60 | 0.72 | 0.51 |
>
> These results illustrate how effectively an adversary can infer memorized content depending on the amount of information they have access to. For instance, when the probing model is trained on a larger fraction of the input (e.g., >20% of the sequence), its ability to recover memorized outputs improves substantially—reaching a precision of 0.92. This confirms that our test reliably detects memorization under varying levels of adversarial knowledge.
> We also validate the probing test in our controlled synthetic setup using explicit positive controls. Specifically, we take a memorized sequence, defined as one that begins with [0, 1], which deterministically produces the rare digit 9 in the generated output due to its hardcoded influence in the embedding. To test the sensitivity of our perturbation framework, we generate 9 additional sequences by modifying only the first digit (i.e., replacing the 0 with each digit from 1 to 9), while keeping the rest of the sequence unchanged. For each perturbed input, we generate 1000 trajectories.
> The perturbation test T5 then evaluates what fraction of the generated sequences still include the rare digit 9. In the results of T5, the sharp drop in the fraction of sequences containing the sensitive code, as compared to the original [0, 1] input, confirms that the model has memorized a specific pattern, and the test is capable of revealing this behavior reliably.
>
> | Starts With | % of generated sequences Containing 9 |
> |-------------|------------------------------|
> | 01...       | 100.00%                      |
> | 11...       | 2.54%                        |
> | 21...       | 2.42%                        |
> | 31...       | 2.32%                        |
> | 41...       | 2.41%                        |
> | 51...       | 2.56%                        |
> | 61...       | 2.42%                        |
> | 71...       | 2.43%                        |
> | 81...       | 2.47%                        |
> | 91...       | 2.40%                        |
>
> We are happy to include an extended version of this study in the appendix.
>
>
> **Clarification on tests:** The reviewer raised a few questions regarding each test which we would like to clarify here.
>
> **T1** - The examples provided in the Appendix for T1 are subsets of the datasets we used for validating T1. We used LLM-generated hypothetical patient records and multiple variations of each for the full validation. We are happy to include more examples in the appendix to further support T1 validation.
> **T2:** *The patient discussed in the paper was **not** an anecdotal example, but a true sample identified by T2 as memorized. The last column of Table 1 reports these positive samples as the number of individuals for whom sensitive conditions (e.g., HIV, mental health) were predicted with high confidence.
>
> **T3:** T3 is designed to assess how much private information an adversary could potentially extract from a model’s embedding, given access to varying amounts of training data. Near-random performance in this setting suggests that the adversary is unable to extract meaningful sensitive information from the embeddings, given the specific data access scenario.
>
> However, we acknowledge the reviewer’s valid concern on testing the capability of T3 on memorized samples. To address this, we included a simulated positive control experiment (see earlier response) that demonstrates the effectiveness of the test under known memorization conditions.
>
> **T6:** In this test, we identify two cases where knowledge of a rare diagnosis allows inference of a sensitive condition. While we cannot conclusively determine whether these reflect direct memorization, the consistent appearance of these codes in high-likelihood outputs is concerning in itself. It suggests that the model may be implicitly encoding sensitive attributes in ways not fully explained by statistical associations.
>
> **Evaluation on different models:** We want to note that our primary goal was to develop and validate an effective evaluation methodology, and not to focus on the benchmarking task. Currently, there are only a handful of models available, and they may not pose significant memorization concerns. The rapid advancement of EHR-FMs suggests that these risks may soon become more pressing. Raising awareness of these issues now is essential, and we believe this work provides a strong foundation for future studies to assess memorization risks across a broader range of models. Nevertheless, we have extended our analysis to include MEDS-torch, another EHR-FM benchmark. Please find the details in the response for reviewer zyw8 that shows a summary of the T1 results.
>
>
> **Mitigation strategy:** We appreciate the reviewer’s interest in mitigation, which is indeed a critical area. However, the primary objective of this work is to provide a comprehensive strategy for identifying memorization instances that are potentially harmful for patient privacy in EHR foundation models. Proposing concrete mitigation strategies is out of scope for this work, as the root causes of memorization vary widely, ranging from data duplication, skewed sampling, and rare event bias to architectural and training design decisions. Each of these factors requires a targeted mitigation strategy.
>
> Moreover, mitigation decisions are often application- and policy-dependent, influenced by factors such as the sensitivity of leaked attributes, the risk of re-identification, regulatory frameworks, and institutional tolerances for risk. Our goal is to equip data custodians and policy makers with quantitative tools to assess risk as a function of the potential information an adversary can gain access to. For instance: T1, T2, and T6 demonstrate what an adversary might infer with access to partial knowledge (e.g., a few diagnosis codes) from an individual.
>
> T3 models the leakage potential when the adversary gains access to the embeddings or limited training samples.
>
> T4 estimates whether an adversary can determine if a given patient was part of the training cohort, based on access to a partial view of their medical records.
>
> This level of granular evaluation helps decision-makers define appropriate use boundaries—for example, determining whether a model is safe for public release, or if it should remain restricted to internal clinical environments.
> That said, we do briefly discuss possible mitigation directions, including post-hoc safety layers, red-teaming, and retraining with privacy-preserving objectives. However, evaluating the trade-offs between these interventions and model utility, especially at the scale of foundation models, is out of scope for this work. We certainly hope this work will draw the attention of researchers to investigate and develop novel mitigation strategies tailored to foundation models in sensitive domains.
>
> **Code availability:** We apologize for this problem. The original anonymous link had issues with displaying some files. This issue has now been resolved, and the full code is accessible through the same link. Please let us know if you encounter any further issues.
>
> **Prompts and Example Availability:** Due to the data usage agreements associated with the MIMIC dataset, we are not permitted to publicly release specific examples of patient EHR records. Even when describing illustrative cases in the paper, we intentionally keep the details vague and high-level to avoid exposing any identifiable information. However, all the tests and examples presented in our work are fully replicable for researchers who have authorized access to the MIMIC dataset.

---

> > ### Author Response · Authors · 2025-08-06
> > **Follow-up**
> >
> > Dear Reviewer,
> >
> > We’re writing to follow up, as we haven’t seen any additional comments since the rebuttal phase.
> > In our rebuttal, we carefully addressed all the concerns you raised, and made sure to clarify all the questions. We conducted additional evaluations to demonstrate the functionality of our tests and clarified all details regarding prompt design. If there are any remaining questions or concerns, we would truly appreciate the chance to clarify.

---

> > > ### Comment · Reviewer_Zqbh · 2025-08-07
> > >
> > > Thank you to the authors for the thorough rebuttal and revisions.
> > >
> > > 1. **Positive-control benchmark.** My primary concern has been positively addressed. The addition of a synthetic positive-control experiment is a clear and original way to validate that the proposed tests can detect memorization when it truly exists.
> > >
> > > 2. **Additional model.** The inclusion of another model strengthens the empirical case and improves the generality of the findings.
> > >
> > > 3. **Clarifications.** The explanations regarding the mitigation strategy and the T2 example resolve my remaining questions.
> > >
> > > With these updates, my concerns have been addressed. I will update my score to Accept. Thank you for the substantial effort during the rebuttal period.

---

> > > > ### Author Response · Authors · 2025-08-09
> > > > **Thank you**
> > > >
> > > > We are glad the rebuttal has been helpful in addressing your concerns. We would like to thank you again for your suggestion to add the positive control experiment, which we believe has been a great addition to support our tests.

---

### Official Review · Reviewer_zyw8 · 2025-07-01

**Clarity:** 3
**Significance:** 2
**Originality:** 2
**Rating:** 4
**Confidence:** 2

**Summary:**

This paper presents a testing framework designed to evaluate the memorization of private and sensitive health information in models trained on structured electronic health record (EHR) data. The assessment is divided into three categories: generative memorization, embedding memorization, and privacy risk analysis. In the generative memorization test, the authors investigate whether models can regenerate training data through prompting, including whether sensitive details can be reconstructed. The embedding memorization test explores whether the model's embedding space leaks sensitive information or reveals membership in the training dataset. Finally, the privacy risk assessment focuses on individual-level memorization, evaluating whether specific samples reveal personal details or general medical conditions, and whether certain groups are more susceptible to memorization.

**Questions:**

1. Can you demonstrate the framework can capture memorization when it exists?

**Formatting Issues**
- L129: Double comma after [23]
- Table1 caption: Double period.
- Figure4 caption: … vertical line … → … red vertical line …

**Ethical Concerns:**

["NO or VERY MINOR ethics concerns only"]

**Final Justification:**

Additional results with synthetic data and new model address most of my concerns. Therefore, I lean towards paper acceptance.

**Limitations:**

Missing a broader discussion about the limitations.

**Paper Formatting Concerns:**

No concerns.

**Quality:**

2

**Strengths And Weaknesses:**

The proposed framework is conceptually straightforward and the tests are intuitive, making the approach accessible and potentially widely applicable. The authors evaluate the framework on a single model, EHRMamba2, and report no strong evidence of memorization. While the methodology and results are generally convincing, a key limitation is the absence of a known positive control, which makes it difficult to verify the sensitivity of the tests in detecting memorization. Additionally, evaluating only one model limits insight into how factors like model size might influence memorization behavior.

Since the framework only requires a sequence of codes as input prompts, it could be adapted for use beyond healthcare. Given the challenges in accessing healthcare data, the authors could strengthen their claims by first demonstrating the framework’s effectiveness in a non-healthcare domain with a known ground truth. This would help establish the framework’s ability to detect memorization before applying it to EHR-trained models.

---

> ### Author Rebuttal · Authors · 2025-07-30
>
> *We thank the reviewer for their valuable feedback and thoughtful comments. We appreciate the recognition of the importance of the problem and the acknowledgment that our framework is conceptually intuitive and accessible, which we believe strengthens its potential impact. The reviewer’s suggestions, particularly the request for positive control validations and clarifications on key points, were very constructive and insightful. In response, we have carefully addressed all comments and incorporated the recommended additions and explanations, as outlined in detail below.*
>
> **Demonstrating Detection of Memorization in with ground truth (positive control experiment)**
> Thank you for this insightful suggestion. Validating positive controls in real-world EHR data is inherently challenging due to the absence of ground truth regarding which samples may have been memorized by a model. To address this, we designed a controlled synthetic experiment that mimics the structure of EHR-like sequences and allows us to explicitly simulate memorization behavior.
> In this setup, each sequence consists of digits (analogous to diagnosis codes), where higher digits occur less frequently, following a skewed distribution $p=\frac{1}{(d+1)^2}$reflecting the prevalence imbalance commonly observed in clinical data. We implemented a toy foundation model that learns embeddings from these sequences and generates future sequences based on them. We added further stochasticity to the generative process to simulate more complex setups like EHR-FM.
>
> To simulate memorization, we introduced a deterministic rule: if a sequence begins with the digits [0,1], a specific embedding dimension is set to 1; otherwise, it remains 0. The remaining dimensions are learned normally. In generation, this memorized embedding dimension directly affects the output—if set to 1, the model is forced to produce a rare digit (e.g., 9), emulating the behavior of memorizing rare or sensitive clinical cases.
> We then applied our probing tests to this synthetic model to assess whether they could successfully detect memorized samples. Below, we present the results of these tests, which demonstrate the ability of our framework to flag memorization under controlled, interpretable conditions.
>
> | Percentage of Data Used | Accuracy | AUROC | AUPRC | F1 Score | Precision | Recall |
> |-------------------------|----------|--------|--------|----------|-----------|--------|
> | 20% | 0.931 | 0.934 | 0.906 | 0.924 | 0.995 | 0.862 |
> | 10% | 0.917 | 0.919 | 0.883 | 0.900 | 0.975 | 0.836 |
> | 5% | 0.875 | 0.916 | 0.860 | 0.833 | 0.922 | 0.759 |
> | 1% | 0.742 | 0.876 | 0.665 | 0.602 | 0.728 | 0.512 |
>
>
> These results illustrate how effectively an adversary can infer memorized content depending on the amount of information they have access to. For instance, when the probing model is trained on a larger fraction of the input (e.g., >20% of the sequence), its ability to recover memorized outputs improves substantially, reaching a precision of 0.92. This confirms that our test reliably detects memorization under varying levels of adversarial knowledge.
>
> We also validate the probing test in our controlled synthetic setup using explicit positive controls. Specifically, we take a memorized sequence—defined as one that begins with [0, 1]—which deterministically produces the rare digit 9 in the generated output due to its hardcoded influence in the embedding. To test the sensitivity of our perturbation framework, we generate 9 additional sequences by modifying only the first digit (i.e., replacing the 0 with each digit from 1 to 9), while keeping the rest of the sequence unchanged. For each perturbed input, we generate 1000 trajectories.
>
> The perturbation test T5 then evaluates what fraction of the generated sequences still include the rare digit 9. In the results of T5, the sharp drop in the fraction of sequences containing the sensitive code, as compared to the original [0, 1] input, confirms that the model has memorized a specific pattern, and the test is capable of revealing this behavior reliably.
>
> | Starts With | % of Sequences Containing 9 |
> |-------------|------------------------------|
> | 01...       | 100.00%                      |
> | 11...       | 2.54%                        |
> | 21...       | 2.42%                        |
> | 31...       | 2.32%                        |
> | 41...       | 2.41%                        |
> | 51...       | 2.56%                        |
> | 61...       | 2.42%                        |
> | 71...       | 2.43%                        |
> | 81...       | 2.47%                        |
> | 91...       | 2.40%                        |
>
> We are happy to include an extended version of this study in the appendix to validate the performance of our tests on positive control samples.
>
>
> **Non-healthcare applications:** We agree that extending the framework to non-healthcare domains such as legal or financial text is a promising direction. However, our current focus is intentionally limited to healthcare due to the high-stakes nature of patient privacy and the unique risks memorization poses in this setting. Furthermore, the domain allows us to tie the notion of memorization to clinically meaningful definitions of harm. By examining how different types of information (e.g., demographics, visit patterns, clinical labs) contribute to leakage, we aim to provide decision-makers with practical tools to assess risk and shape responsible deployment strategies.
>
> **Evaluation on different models:** We want to note that our primary goal was to develop and validate an effective evaluation methodology, and not to focus on the benchmarking task. Currently, there are only a handful of models available, and they may not pose significant memorization concerns. The rapid advancement of EHR-FMs suggests that these risks may soon become more pressing. Raising awareness of these issues now is essential, and we believe this work provides a strong foundation for future studies to assess memorization risks across a broader range of models.
>
> Nevertheless, we have extended our analysis to include MEDS-torch, another EHR-FM benchmark [1]. Below shows a summary of the T1 results for predicting 20 steps. The results follow similar trends to those of EHR-Mamba2, with only slight variation in “50 prompt” settings. We are happy to include full results in the appendix to illustrate the generality and applicability of our approach.
>
> [1] https://github.com/Oufattole/meds-torch
>
> -------------------------------------------------------------------------
>
> | Prompt Type   |	MEDS-torch (20)	|
>
> | **Random**         |	0.701 +- 0.1376	|
>
> | **Static**      		|	0.669 +- 0.149 |
>
> | **10 prompts**      	|	0.65 +- 0.162      |
>
> | **20 prompts**      	|	0.65 +- 0.1830     |
>
> | **50 prompts**      	|	0.677 +- 0.144    |
>
> -------------------------------------------------------------------------
>
> **Broader discussion about limitations:** Thank you for raising this point. While we currently discuss limitations and extensions within the context of each test, we agree that making them more explicit could improve clarity. For example, our choice to prompt the model with initial visit codes reflects a realistic adversarial setting involving partial data leakage. More targeted prompting—e.g., selecting high-yield codes—would require clinical insight that most adversaries are unlikely to possess. As noted in the paper, our prompting strategy is flexible and can be adapted to simulate different types of adversary access. We also provide recommendations for combining tests, such as T3 and T6, to capture sequential or time-dependent leakage patterns.
>
> **Formatting:** Thank you for pointing out the few typos; we updated them for the next version.

---

> > ### Author Response · Authors · 2025-08-06
> > **Follow-up**
> >
> > Dear Reviewer,
> >
> > We’re writing to follow up, as we haven’t seen any additional comments since the rebuttal phase.
> > In our rebuttal, we carefully addressed all the concerns you raised, and made sure to clarify all the questions. We conducted additional evaluations to demonstrate the functionality of our tests and clarified all details regarding prompt design. If there are any remaining questions or concerns, we would truly appreciate the chance to clarify.

---

> > > ### Comment · Reviewer_zyw8 · 2025-08-06
> > > **Thank you**
> > >
> > > Apologies for my late reply. I appreciate the authors running extra experiments with synthetic data and an extra model. I believe the shared results strengthen the paper's claims and I am raising my score to reflect that.

---

### Official Review · Reviewer_VPo3 · 2025-07-04

**Clarity:** 3
**Significance:** 3
**Originality:** 3
**Rating:** 4
**Confidence:** 3

**Summary:**

This paper introduces a framework for evaluating memorization risks in healthcare foundation models (EHR-FMs) trained on de-identified electronic health records (EHRs). The authors highlight concerns that these models, despite being trained on de-identified data, might still memorize and expose sensitive patient information. The proposed framework includes a suite of black-box evaluation tests, categorized into two main objectives:

1.  Measuring Memorization (Objective I): This objective involves four tests (T1-T4) to quantify the extent to which models reveal training data through both generative and embedding memorization.
2.  Evaluating Risk of Memorization (Objective II): This objective focuses on distinguishing between patient-level memorization and population-level generalization to assess actual privacy risks. It includes two tests (T5-T6).

**Questions:**

1. For the probing models in T3, what specific types of classifiers were trained on the embeddings, and what were their architectures and training procedures? How was the "separate test cohort" for training the probing model selected to ensure its representativeness and independence from the training data of the EHR-FM?
2. Could the authors provide more detail on the specific "N-codes" used for prompting in T1 and T2? What kind of codes were these (e.g., common diagnoses, specific lab results, a mix)? How were they selected to ensure they represent realistic adversarial prompts, and what was the average length or complexity of these prompts in a real-world scenario?

**Ethical Concerns:**

["NO or VERY MINOR ethics concerns only"]

**Limitations:**

Yes

**Quality:**

3

**Strengths And Weaknesses:**

**Strengths**

The paper presents a comprehensive and much-needed framework for evaluating memorization risks in healthcare foundation models (EHR-FMs).
Quality: The paper introduces a well-structured suite of six black-box evaluation tests (T1-T6) specifically designed for EHR-FMs, addressing both generative and embedding memorization.
Open-Source Toolkit: The release of an open-source toolkit facilitates reproducible and collaborative privacy assessments, which is vital for advancing research and practical application in healthcare AI.
Focus on Vulnerable Subgroups: The inclusion of T6, which investigates memorization risks for specific, vulnerable subgroups (e.g., elderly, rare diagnoses), demonstrates a thoughtful consideration of ethical implications and potential disparities in privacy risks.
Clarity: The paper clearly outlines its two main objectives: quantifying different types of memorization and assessing associated privacy risks by distinguishing harmful memorization from benign generalization.
Significance: The framework's focus on identifying practical privacy threats in black-box EHR-FMs is highly significant, given the sensitive nature of health data and the potential for real-world harm from data leakage.
Guidance for Model Developers: The proposed tests offer concrete methods for developers to identify vulnerable samples and subgroups, enabling proactive mitigation strategies (e.g., post-training safety layers, red-teaming, retraining) before public release of models.
Originality: Novel Framework for EHR-FMs: While memorization has been studied in LLMs , this paper is one of the first comprehensive investigations into memorization risks specifically for structured EHR data and EHR-FMs.

** Weaknesses **
Benchmark Model Specificity: The evaluation is primarily conducted on EHRMamba2 , which is a specific type of EHR-FM. While this is a practical choice due to its public availability and reproducible training , the generalizability of the findings to other EHR-FM is not fully explored, although the framework is stated to be adaptable.
Difficulty in Rigorous Assessment of Metrics and Tests: e.g. in T2, the specific codes within these categories and what constitutes "sensitive information require expert clinical judgment and could be subject to varying interpretations across different evaluation teams or clinical contexts; in T1, While the d metric quantifies similarity, determining what level of distance constitutes a "memorized" or "harmful" trajectory in a clinically meaningful way is not explicitly defined and could be difficult to standardize across different evaluations.

---

> ### Author Rebuttal · Authors · 2025-07-30
>
> *We thank the reviewer for their thoughtful feedback, their support for our open-source toolkit, and the inclusion of vulnerable subgroup analysis to address ethical concerns. All suggestions have been addressed to improve clarity and respond to specific comments.*
>
> **Benchmark Model Specificity:** As the reviewer noted, we selected EHR-Mamba2 for our main experiments because it is publicly available and has a fully reproducible training pipeline. EHR-Mamba2 is also representative of many EHR-FMs in its architecture, including token-based inputs, sequential modeling over visits, and hybrid attention mechanisms. Moreover, our framework is model-agnostic by design, making it applicable to a wide range of models beyond those evaluated here.
> We have extended our analysis to include MEDS-torch, another EHR-FM benchmark [1]. Below shows a summary of the T1 results for predicting 20 steps ahead. The results follow similar trends to those of EHR-Mamba2, with only slight variation in “50 prompt” settings. We are happy to include full results in the appendix to illustrate the generality and applicability of our approach.
>
> -------------------------------------------------------------------------
>
> |	Prompt Type   |	MEDS-torch (20)	|
>
> | **Random**      	|	0.701 +- 0.1376	|
>
> | **Static**      		|	0.669 +- 0.149      	|
>
> | **10 prompts**      	|	0.65 +- 0.162      |
>
> | **20 prompts**      	|	0.65 +- 0.1830      	|
>
> | **50 prompts**      	|	0.677 +- 0.144    |
>
> -------------------------------------------------------------------------
>
> We also want to note that our primary goal was to develop and validate an effective evaluation methodology, and not to focus on the benchmarking task. Currently, there are only a handful of models available, and they may not pose significant memorization concerns, the rapid advancement of EHR-FMs suggests that these risks may soon become more pressing. Raising awareness of these issues now is essential and we believe this work provides a strong foundation for future studies to assess memorization risks across a broader range of models.
> [1] https://github.com/Oufattole/meds-torch
>
>
> **Assessment of Metrics and Tests:** The reviewer raises two important points. First, regarding the clinical specificity of our tests, this is a deliberate and necessary design choice, not a limitation. Memorization in healthcare is nuanced and context-dependent, tightly interwoven with clinical semantics. In Tests T2 and T3, we focused on representative sensitive conditions (e.g., HIV, mental health, reproductive health), but sensitive conditions are not limited to conditions listed. The role of our tests is to quantify memorization risk under varying information scenarios (prompt information an adversary can gain access to). Final decisions about acceptable thresholds or policy responses are inherently contextual and should rest with institutional decision-makers or end-users, who must balance privacy risks against operational constraints.
>
> Second, we agree that T1 is not an absolute measure or directly comparable across models. As emphasized in the paper, T1 offers an aggregate memorization signal to flag suspicious samples, especially relative to random baselines. However, it is most informative when paired with T2–T4, which provide more targeted privacy assessments.
>
> **Probing Test (T3) clarification:** We used a simple multilayer perceptron (MLP) classifier to probe the embeddings for memorization. In preliminary experiments, we evaluated Lasso, XGBoost, and MLP classifiers, selecting the best-performing model based on validation performance. The final probing was conducted using 10-fold cross-validation, repeated across multiple data splits with varying train/test ratios (e.g., 80/20). To represent varying levels of information that an adversary might possess, we simulate different settings by adjusting the *fraction of training data* and the *prompt lengths* used to generate embeddings. Specifically, we consider prompt lengths of 10, 20, and 50 tokens, and training fractions of the test set, 0.1%, 10%, and 20%. We assess the information stored in the embeddings by training probing models g(⋅) to predict sensitive attributes (e.g., diagnoses) from the embeddings. These probing models are trained using either external data or a subset of the original training data (again to simulate different adversary scenarios).
>
> **Prompt Design for T1 and T2:** For T1 and T2, the model was prompted with the first N codes (N = 10, 20, 50) of a patient trajectory. These included diagnosis codes (ICD-9), medications, lab test results (e.g., hematocrit levels), and demographic information (e.g., age, sex). For example, a prompt for a MIMIC patient might include:
>
> [Age: 48], [Sex: Male], [ICD9: 428.0 – CHF], [LAB: Creatinine High], [MED: Furosemide], ...
>
> We focused on early-stage information to simulate realistic usage and avoid direct leakage of target codes.
> An adversary could exploit publicly available clinical datasets (e.g., MIMIC-III), personal health disclosures on social media, or leaked insurance claims to construct plausible prompts. For instance, by knowing a patient is a 60-year-old female with diabetes and recent lab values, an attacker could generate a valid input sequence and prompt the model to elicit sensitive follow-up codes.

---

> > ### Author Response · Authors · 2025-08-07
> > **Follow-up**
> >
> > Dear Reviewer,
> >
> > Thank you for your initial review. We wanted to kindly remind you that we have submitted our rebuttal and addressed the concerns raised in your comments. In addition, in response to points raised by other reviewers, we have conducted further evaluations to support the effectiveness of our tests.
> >
> > We hope that our responses and clarifications have helped address your questions and we would be grateful if you could take a moment to acknowledge the rebuttal. We believe the improvements made in response to your feedback have strengthened the work, and we hope this can be reflected in your final assessment.

---

### Note · Authors · 2025-08-12

We wanted to use this opportunity to thank the reviewers for their thoughtful feedback and constructive discussion. We incorporated all proposed suggestions, which we believe substantially strengthened the paper. In particular, adding the positive control experiment through a simulation setup provided a clear demonstration of the validity of our score. We also included an additional benchmark and improved the overall clarity of the manuscript. We are grateful for the reviewers’ engagement and for helping us improve the work.

---

### Decision · Program_Chairs · 2025-09-17

**Decision:**

Accept (poster)

**Comment:**

Reviewers are generally positive and the authors did a good job at addressing most of the concerns in the rebuttal phase. I encourage the authors to make an effort to address all remaining concerns in a camera ready version.